# OpenReview forum: "LLM Unlearning via Calibrated and Tokenized Negative Preference Alignment"
_ICLR.cc/2026/Conference — Submitted to ICLR 2026_

### Official Review · Reviewer_vLqU · 2025-10-29

**Soundness:** 3
**Presentation:** 3
**Contribution:** 3
**Rating:** 6
**Confidence:** 3

**Summary:**

The paper focuses on LLM unlearning—selectively erasing undesirable knowledge such as hazardous procedures or copyrighted content while preserving general capabilities. Traditional GA methods often cause catastrophic forgetting and require retention data to maintain performance. NPO improves on this but remains limited by static reference models and length-biased gradients. The authors introduce CATNIP, a calibrated and tokenized negative preference alignment method that uses a reverse policy (1 − πθ) as an adaptive reference to scale unlearning strength with model confidence, and applies token-level optimization to eliminate sequence length bias. This enables precise, data-efficient unlearning without needing retention datasets or contrastive pairs. The approach is theoretically grounded in policy ranking via the Bradley-Terry model and empirically validated on hazardous knowledge (WMDP) and copyrighted text (MUSE-Harry Potter) tasks.

**Strengths:**

The paper stands out for its rigorous theoretical foundation, deriving the method from preference optimization principles and providing clear gradient formulations that explain why it outperforms GA or NPO, backed by ablation studies that isolate components like tokenization and calibration.

**Weaknesses:**

- Use of un-normalized reverse policy (1 − πθ) as reference risks instability when model confidence is low or distributions are multimodal; no discussion of failure modes or mitigation.
- Evaluation limited to two benchmarks and smaller models; lacks comparison to recent retention-free methods or large-scale 7B+ models.
- Data efficiency claims rely on 132 QA pairs but include no scaling analysis below this threshold or under noisy/real-world unlearning requests.

**Questions:**

- Why not explore learned or EMA-based reference policies instead of the fixed reverse approximation?
- Any plans to evaluate on larger models or more diverse unlearning targets?
- How does CATNIP differ from [1] (which mentions calibration) and [2] (which mentions tokenization)—do they share core ideas or are the approaches fundamentally distinct?

[1] Wang, Qizhou, et al. "Towards effective evaluations and comparisons for llm unlearning methods." arXiv preprint arXiv:2406.09179 (2024).

[2] Yang, Puning, et al. "Exploring Criteria of Loss Reweighting to Enhance LLM Unlearning." arXiv preprint arXiv:2505.11953 (2025).

---

> ### Author Response · Authors · 2025-11-24
> **Response to Reviewer vLqU**
>
> Dear reviewer vLqU,
>
> Thank you for your thoughtful review and positive evaluation of our work. We greatly appreciate your recognition of our theoretical grounding, the clarity of our gradient formulations, and the effectiveness of our ablation studies. Please find our detailed responses below to your comments.
>
> ---
>
> **Weakness 1.1:** Use of un-normalized reverse policy as reference risks instability when model confidence is low or distributions
> **Response**: Thank you for your thoughtful comment.  Actually, our choice of reference $1 - \pi_\theta$ can smooth the confidence weight and stablizes training:
>
> - As shown in Figure 1 of the paper, given $\beta >= 1$ (In our experiments, we always used $\beta >=1$), the  weight $w_i(\beta, \pi)$ derived by this reference choice is bounded. Especially, when $\beta>1$, the weight function gets a smooth curve when the current policy $\pi_\theta$ is over-confident in generating or refusing  a token( *i.e.*  $\pi_\theta \to0$ or $\pi_\theta \to 1$).
> - When $\pi_\theta$ reaches maximum uncertainty ( $\pi_\theta \to 0.5$), the weight derivative reaches the highest slope, which can appropriately amplify the learning signal.
>
>
> **Weakness 1.2**: No discussion of failure modes or mitigation.
> **Response**: Thank you for raising this important point. We consider two mitigation techniques: 1) The formulation of CATNIP can serve as a building block and be combined with engineering efforts such as gradient clipping to further stablize training. 2) Addtionally, although CATNIP is designed to work without retention data, it can be augmented with a retention regularization loss when such data is available. To verify this, we conducted experiments combining CATNIP with a KL retention loss (similar to GA + KL), with results summarized in  **Table 1** below.  CATNIP + KL achives the best tradeoff in unlearning and utility preservation. In contrast, GA + KL largely hurts unlearning efficacy, while removing the KL retention term causes catastrophic forgetting. This indicates CATNIP exhibits minimal interference between unlearning and retention objectives, and demonstrates higher compatibility with potential risk mitigation techniques.
>
> ---
>
> **Table1:** Evaluating CATNIP with retention loss incoporated as mitigation.
>
> | Method                            | Know f ↓ (Extended) | Δf ↓ (Extended) | Know f ↓ (MUSE) | Δf ↓ (MUSE) | MMLU ↑    | Δu ↑ | ΔO ↑      |
> | --------------------------------- | ------------------- | --------------- | --------------- | ----------- | --------- | ---- | --------- |
> | **Base model**                    | 39.99               | -               | 32.13           | -           | 60.45     | -    | -         |
> | GA + KL (w/ $D_r$)                | 38.29               | (✗)             | 27.20           | (✓)         | **60.18** | (✓)  | 1.43      |
> | NPO + KL (w/ $D_r$)               | 33.62               | (✗)             | 28.92           | (✗)         | 59.47     | (✓)  | 5.39      |
> | FLAT (w/ $D_{ct}$)                | 5.44                | (✓)             | 6.35            | (✓)         | 50.12     | (✗)  | 24.22     |
> | **CATNIP (Ours) + KL (w/ $D_r$)** | **0.00**            | (✓)             | **0.00**        | (✓)         | **59.48**      | (✓)  | **39.02** |
> | GA                                | **0.00**            | (✓)             | **0.00**        | (✓)         | 24.87     | (✗)  | -5.61     |
> | NPO                               | 25.21               | (✗)             | 24.18           | (✗)         | 54.79     | (✓)  | 9.12      |
> | SimNPO                            | 6.87                | (✓)             | 6.54            | (✓)         | 51.84     | (✓)  | 24.21     |
> | **CATNIP (Ours)**                 | **2.29**            | (✓)             | **2.08**        | (✓)         | 52.17     | (✓)  | **29.42** |

---

> ### Author Response · Authors · 2025-11-24
> **Response to Reviewer vLqU**
>
> **Weakness2: Evaluation limited to two benchmarks and smaller models; lacks comparison to recent retention-free methods.**
>
> **Response:** We appreciate this constructive comment. We have added new experiments that include direct comparisons against recent unlearning methods such as WGA [1] and SatImp [2], which can be applied in a *retention-data-free* setting and therefore constitute fair and meaningful baselines for comparison with CATNIP.
>
> Comparing to WGA [1] and SatImp [2], our method remains competitive on realistic unlearnining tasks including WMDP-Cyber, WMDP-Bio, and MUSE-Book. As summarized in Table 2 & 3 below, prior work do not consistently maintain performance across realistic conceptual unlearning tasks:
>
> - WGA matches CATNIP on MUSE-Book forgetting performance but exhibits notable over-forgetting of general knowledge (MMLU) on WMDP-Cyber and WMDP-Bio tasks.
> - SatImp performs well on WMDP-Cyber but shows catastrophic forgetting on WMDP-Bio and underperforms CATNIP on MMLU within the MUSE-Book setting.
>
> In contrast, CATNIP performs well across  three tasks, further supporting the reliability and robustness of our method.
>
> **Table2: WMDP Performance Comparison.**
>
> | Methods                     | Bio ↓ | Δf ↓ | MMLU↑ | Δu ↑ | ΔO ↑ | Cyber ↓ | Δf ↓ | MMLU↑ | Δu ↑ | ΔO ↑ |
> |-----------------------------|-------|------|--------|-------|--------|----------|------|--------|-------|--------|
> | **Base model**              | 63.70 | -    | 58.10  | -     | -      | 44.00    | -    | 58.10  | -     | -      |
> | RMU (w/ $D_{retain}$)            | 31.89 | (✓)  | 57.18  | (✓)   | 30.89  | 26.93    | (✓)  | 57.81  | (✓)   | 16.78  |
> | GA + KL (w/ $D_{retain}$)        | 62.77 | (✗)  | 57.29  | (✓)   | 0.12   | 40.36    | (✗)  | 59.82  | (✓)   | 5.36   |
> | NPO + KL (w/ $D_{retain}$)       | 63.16 | (✗)  | 57.67  | (✓)   | 0.11   | 39.61    | (✗)  | 57.11  | (✗)   | 3.30   |
> | FLAT (w/ $D_{ct}$)               | 25.61 | (✓)  | 27.16  | (✗)   | 7.15   | 24.51    | (✓)  | 23.24  | (✗)   | -15.37 |
> | RMU*                        | 25.84 | (✓)  | 25.50  | (✗)   | 5.26   | 24.61  | (✓)  | 25.50  | (✗)   | -13.21 |
> | GA                          | 24.65 | (✓) | 25.25 | (✗)   | 2.60   | 33.77    | (✗)  | 48.79  | (✓)   | 0.92   |
> | NPO                         | 62.69 | (✗)  | 56.88 | (✓) | -0.21 | 36.89    | (✗)  | 55.34 | (✓) | 4.35   |
> | SimNPO                      | 27.10 | (✓)  | 47.37  | (✗)   | 25.87  | 34.22    | (✗)  | 54.25  | (✓)   | 5.93   |
> | WGA                         | 24.59 | (✓)  | 23.31  | (✗)   | 4.30   | 26.07    | (✓)  | 41.30  | (✗)   | 1.13   |
> | SatImp                      | 24.27 | (✓)  | 26.27  | (✗)   | 7.60   | 29.79    | (✓)  | 52.99  | (✓)   | 9.10   |
> | **CATNIP (Ours)**           | 28.36 | (✓) | 51.37 | (✓) | 28.61 | 28.69 | (✓) | 53.01 | (✓) | **10.22** |
>
> **Table3: MUSE-Book Performance Comparison.**
>
> | Method                       | Know f ↓ (Extended) | Δf ↓ (Extended) | Know f ↓ (MUSE) | Δf ↓ (MUSE) | MMLU ↑ | Δu ↑ | ΔO ↑ |
> |------------------------------|---------------------|------------------|------------------|--------------|--------|-------|-------|
> | **Base model**               | 39.99               | -                | 32.13            | -            | 60.45  | -     | -     |
> | GA + KL (w/ D_r)             | 38.29               | (✗)              | 27.20            | (✗)          | 60.18 | (✓)     | 1.43  |
> | NPO + KL (w/ D_r)            | 33.62               | (✗)              | 28.92            | (✗)          | 59.47  | (✓)     | 5.39  |
> | FLAT (w/ D_ct)               | 5.44                | (✓)              | 6.35             | (✓)          | 50.12  | (✗)   | 24.22 |    |
> | GA                           | 0.00           | (✓)              |     0.00         | (✓)          | 24.87  | (✗)     | -5.61 |
> | NPO                          | 25.21               | (✗)              | 24.18            | (✗)          | 54.79  | (✓)  | 9.12  |
> | SimNPO                       | 6.87                | (✓)              | 6.54             | (✓)          | 51.84  | (✓)     | 24.21 |
> | WGA                          | 2.09                | (✓)              | 3.25             | (✓)          | 40.24  | (✗)     | 27.85 |
> | SatImp                       | 0.00            | (✓)              | 0.00         | (✓)          | 41.84  | (✗)     | 21.38 |
> | **CATNIP (Ours)**            | 2.29            | (✓)              |  2.08        | (✓)          | 52.17  | (✓)     | **29.42** |
>
> ###
>
> [1] Rethinking LLM Unlearning Objectives: A Gradient Perspective and Go Beyond. ICLR2025.\
> [2] Exploring Criteria of Loss Reweighting to Enhance LLM Unlearning. ICML2025.

---

> ### Author Response · Authors · 2025-11-24
> **Response to Reviewer vLqU**
>
> **Weakness3:** Data efficiency claims rely on 132 QA pairs.
>
> **Response:** Thank you for this thoughtful observation. We would like to clarify that the 132 **training** quereis are designed as a controlled proof-of-concept to demonstrate CATNIP's data efficiency, specifically, its ability to achieve concept forgetting with minimal supervision.
>
> We agree that real-world unlearning scenarios involve noisier, larger-scale data. To address this, our **evaluations** use the MUSE-Book benchmark (with extended queries beyond those originally provided) and WMDP benchmark, both involving large volumes of text that reflect practical unlearning complexity. Together, these experiments provide evidence of CATNIP's effectiveness across different realistic settings.
>
> ---
>
> **Question:** Why not explore learned or EMA-based reference policies instead of the fixed reverse approximation?
>
> **Response**: Thank you for this insightful question.  Our choice of $1 - \pi_\theta$ arise from its effectiveness in meaningful reference guidane, and its computational and data efficiency:
>
> **(1)** During unlearning, $1 - \pi_\theta$ is **automatically adapted** along with the update of $\pi_\theta$, and **immediately available**, which **naturally derives** a token weight $w_i$​ (Eq 9.) There is no need to manually craft a confidence weight as other work does, or storing a static reference from a frozen model copy,  thus saving more runtime memory.
>
> **(2)** It effectively forms  a *counterfactual*  guidance from the start. In the initial unlearning stage, the target policy $\pi_\theta$ is prone to generating forgetting knowledge. Using $1-\pi_\theta$ is more likely to refuse to generate such content,  providing a strong initial guidance. In contrast, $\pi_\text{ref} = \pi_\theta$ serves as a weaker reference when training starts, as it mirrors rather than counters the models' undesirable behavior.
>
> ---
>
> **Question:** How does CATNIP differ from [1] (which mentions calibration) and [2] (which mentions tokenization)—do they share core ideas or are the approaches fundamentally distinct?
>
> **Response:** Thank you for pointing us to other recent work. We would like to clarify CATNIP and [1] use calibration in two orthogonal senses. In CATNIP, calibration refers to the gradient-level calibration of the unlearning objective in the training stage, we applied reverse policy $1-\pi_{\theta}$ to calibrate the gradient of each token according to the model’s confidence. [2] use calibration purely on the evaluation side. They align LLM performance on non-targeted data post-unlearning to ensure an easy and fair way of comparison by model mixing: $(1-\alpha)\pi_{ref}+\alpha \pi_{\theta}$, where $\pi_{ref}$ denotes policy before unlearning and $\pi_{\theta}$ denotes policy after unlearning. By careful-adjusted $\alpha$, they can control the extent of unlearning to align performance on non-targeted data, similar to those before unlearning.
>
> SatImp [2] proposed a token-wise unlearning method inspired by two empirical concepts: Saturation and importance, while our formulation provides a unified and principled theoretical framework. Our choice of using a reverse policy as the reference naturally leads to a **principled** gradient rescaling (Eq. (8)–(9)) rather than  heuristically  crafting a reweighting coefficient. It effectively addresses the *saturation* issue discussed in [2]. Subsequently, we incorporated a token-wise loss to reflect such rescaling on fine-grained tokens rather than entire sequence  - which echoes the *importance* concept in [2] while being grounded in our policy-preference formulation.
>
> Comparative experiments (please refer to Table 2) above show that, SatImp performs well on WMDP-Cyber but shows catastrophic forgetting on WMDP-Bio and underperforms CATNIP on MMLU within the MUSE-Book setting. In contrast, CATNIP performs well across  three tasks.
>
>
> [1] Wang, Qizhou, et al. "Towards effective evaluations and comparisons for llm unlearning methods." arXiv preprint arXiv:2406.09179 (2024).
>
> [2] Yang, Puning, et al. "Exploring Criteria of Loss Reweighting to Enhance LLM Unlearning." arXiv preprint arXiv:2505.11953 (2025).
>
>
> ---
> We are running an additional experiment on 7B+-scale LLMs to further validate the generality of our findings. Due to computational cost, this experiment is still in progress. Importantly, all existing evidence presented in the rebuttal has supported our conclusions, and the preliminary results we have obtained so far are consistent with the reported trends.
>
> Once again, we sincerely appreciate you for the invaluable feedback and constructive questions. We look forward to any further feedback you may have.
>
> Sincerely,\
> Authors

---

### Official Review · Reviewer_prd3 · 2025-10-30

**Soundness:** 2
**Presentation:** 3
**Contribution:** 1
**Rating:** 2
**Confidence:** 5

**Summary:**

This paper aims to address LLM unlearning challenges with two perspective: token-wise loss fuction design and real-time reference model optimization. While the topic is clear and easy to understand, the overall motivation and contribution of the work are also clearly articulated but it is not novel. The paper does not convincingly demonstrate why the proposed approach is necessary or how it significantly differs from existing methods.

From a methodological perspective, the proposed idea appears to be a relatively minor variation of prior work. Although this paper compare with several classicial methods in LLM unlearning, recent progresses in ICLR2025 and ICML2025 are ignored which are significantly related tothe contributions of this paper. Thus, the novelty is limited, and the technical contribution lacks depth. More details about this evaluation will be presented in weaknesses.

While the presented experiments provide some evidence of the proposed method’s effectiveness, the claimed superiority and general applicability remain questionable. Consequently, the analysis and conclusion — particularly the statement that the method achieves “stronger knowledge forgetting and preservation trade-offs than state-of-the-art methods” — are not fully convincing.

The paper is well written, with clear exposition and logical organization. However, despite the good writing quality, the major weaknesses in motivation, methodology, and experimental validation significantly limit the overall contribution. Clear writing alone cannot compensate for these substantive shortcomings, and therefore cannot serve as a sufficient reason for acceptance.

**Strengths:**

The motivation of the paper and the corresponding methodological design are centered around two main observations:

1. Sample-wise optimization limitation: Existing unlearning methods operate in a sample-wise manner, without distinguishing between short and long answers. As a result, longer samples receive disproportionate optimization attention, potentially biasing the learning process.

2. Static reference model issue: Current approaches rely on a static reference model (i.e., the pre-unlearned model). During training, some samples are densely distributed in the static model’s feature space, which limits the effectiveness of unlearning for those samples and prevents the model from fully leveraging the knowledge they contain.

These issues are indeed timely and important for LLM unlearning. The paper clearly identifies them and attempts to propose a “novel” method to address them. Overall, the manuscript is well organized and clearly written.

**Weaknesses:**

1. Motivation: As mentioned in the Strengths section, the paper’s motivation focuses on two subtle aspects of classical unlearning methods. These aspects are indeed relevant and could, in principle, inspire meaningful improvements in future work. However, it is unfortunate that the authors seem largely unaware of recent advances in this area.

    Specifically, both of the paper’s stated motivations have already been explored in the recent literature.

    (1). Token-wise optimization: Several works published at ICLR 2025 [1] and ICML 2025 [2] have already introduced token-wise loss designs for LLM unlearning tasks, addressing the same limitation discussed here. Thus, the first motivation lacks novelty.

    (2). Dynamic reference models: Similarly, existing token-wise approaches [1][2][3] already adopt dynamic reference models ($\phi_t$) instead of static ones ($\phi_0$). Consequently, the second motivation also fails to offer a genuinely new perspective.

    In summary, while the motivation is clearly articulated, it does not present any substantial innovation compared to recent state-of-the-art research.


2. Method: As shown in Equation (9), the proposed CaTNiP essentially functions as a token-wise reweighting method. Prior studies have already demonstrated empirically that token-wise approaches generally outperform sample-wise ones.

    However, this paper does not include comparisons with other token-wise baselines (e.g., WGA[1], SatImp[2]), yet still claims to outperform the state of the art. Such a claim is not well supported and therefore lacks credibility.

3. Experiment: The experimental section is insufficient to validate the proposed method.

     (1). Benchmark: Only the WMDP and part of the MUSE benchmarks are utilized, leaving out other important datasets such as TOFU.

     (2). Model: Given that Zephyr-7B is a standard backbone for WMDP, the choice is reasonable; however, the rest of the experiments are conducted solely with LLaMA-3B. This setup is too limited to demonstrate the generality of CaTNiP.

    It remains unclear how the method performs on smaller (e.g., 1B) or larger (e.g., 8B) models. Similarly, results on MUSE-News are missing—only MUSE-Books is reported. Furthermore, TOFU is one of the most widely adopted benchmarks for LLM unlearning, yet no experiments are provided on it. The absence of TOFU results significantly weakens the empirical validation and makes it difficult to assess the robustness and general applicability of the proposed approach.

[1] Rethinking LLM Unlearning Objectives: A Gradient Perspective and Go Beyond. ICLR2025

[2] Exploring Criteria of Loss Reweighting to Enhance LLM Unlearning. ICML2025

[3] RULE: Reinforcement UnLEarning Achieves Forget–retain Pareto Optimality. Arxiv 2506.07171, NIPS2025

**Questions:**

Please refer to the weakness.

---

> ### Author Response · Authors · 2025-11-24
> **Response to Reviewer prd3 (1/4)**
>
> Dear Reviewer prd3,
>
> Thank you for your thoughtful review. We appreciate the reviewer's acknowledgment of the paper’s clear motivation and overall organization. We sincerely thank the reviewer for pointing us to relevant recent work [1, 2, 3]. We have now thoroughly compared with prior work with both discussions and updated experiments in the manuscript. Please find below our responses to your comments.
>
>
> **Weakness 1: Regarding motivation and novelty compared with prior work**:
>
> While we acknowledge and appreciate the contributions of recent works [1,2]  in LLM unlearning by exploring token-wise objectives, our method differs fundamentally in both theoretical framing and formulation:
>
> 1. **Theoretical foundation:**  Our work proposed the theoretical framework that treats unlearning as preference optimization over pairs of ***policies*** rather than pairs of ***data samples*** (Sec 3.1). This removes a theoretical limitation of NPO, which inherits from DPO but empirically ignores the loss component related to the preferred data sample (Section 2.1).
> 2. Based on this *preference-over-policy-pair* concept, we  proposed to use a *reverse policy* as a conterfactual reference, which naturally leads to a **principled** gradient rescaling (Eq. (8)–(9)) rather than  heuristically  crafting a reweighting coefficient. It effectively addresses the *saturation* issue discussed in [2]. Subsequently, we incorporated a token-wise loss to reflect such rescaling on fine-grained tokens rather than the entire sequence, which echoes the *importance* concept in [2] while being grounded in our policy-preference formulation. We further showed that the combination of reverse policy calibration + tokenization is essential via ablations (Table 3 in the Paper).
> 3. Given this formulation, our method is designed to fit well in a retention-data-free setting that solely utilizes forgetting data, unlike prior work that requires crafting a boundary dataset [3]. In the meantime, when retention data is available, our method can inoporate a retention loss to further improve unlearning and utility preservation performance on the MUSE-Book task (e.g. Table 3 in response to Reviewer agwH).
>
> We would like to clarify that while our method benefits from token-wise loss, similar to [1,2], the key difference lies in its derivation: [1,2]     focus on designing reweighting strategies to address the limitations of GA. Meanwhile, the weight $w_i$ in our method is naturally derived and is rooted in the above policy preference optimization. We acknowledge the wisdom of using token-wise loss in LLM post-training, which has been adopted by other work such as RL-based alignment (e.g., DAPO compared to DRPO). This does not diminish the contribution of methods that operate on a token-wise loss but address different conceptual challenges.
>
> ---
>
> **Weaknesses 2 & 3: Did not include comparisons with other token-wise baselines on more benchmarks**.
>
> **Response:** We thank the reviewer for raising this important point. We have added new experiments that include direct comparisons against token-wise baselines such as WGA [1] and SatImp [2]. These methods can be applied in a *retention-data-free* setting and therefore constitute fair and meaningful baselines for comparison with CATNIP.
>
> **Clarification on Benchmark Choices**: We would like to respectfully clarify the rationale behind our initial selection of benchmarks:
>
> 1. While **TOFU** is a meaningful benchmark for unlearning research, the forgetting dataset and retention dataset are **synthesized** and are **(near-)IID** distributed. In our personal opinion, this might fully reflect our targeted unlearning scenario: **realistic conceptual unlearning**, where **i)** the forgotten knowledge should be realistic and exist in the base LLM, and **ii)** the retained concept should be semantically distinct from the forgotten one. Resultingly, the model utility (MU) metric in the TOFU benchmark is a harmonic mean of 9 utility metrics, including performance on the synthetic retention domain, which does not directly reflect the conceptual-level separation we aim to achieve. Hence, we initially prioritized benchmarks whose tasks involve realistic, pre-existing knowledge already existing in foundation models.
> 2. Existing unlearning methods have reported combinations of benchmarks based on their research problem focus and settings. For instance, the paper [3], as mentioned by the reviewer, designs boundary data for refusal-style unlearning yet does not evaluate on TOFU, as TOFU’s synthetic forgetting and retention datasets are not easily separable at a semantic level. Similarly, WGA [1] does not evaluate on WMDP, which was introduced after TOFU and targets a different category of unlearning tasks.

---

> ### Author Response · Authors · 2025-11-24
> **Response to Reviewer prd3 (2/4)**
>
> As per the reviewer's suggestions, we have now included comprehensive comparisons on both:
> - Realistic unlearning tasks: WMDP-Cyber, WMDP-Bio, MUSE-Book
> - Synthetic unlearning tasks (with near-IID forgetting and retention data): TOFU.
>
> Comparing to WGA [1] and SatImp [2], our method remains competitive on ***realistic unlearning tasks*** including WMDP-Cyber, WMDP-Bio, and MUSE-Book. For ***synthetic unlearning tasks*** (TOFU) with **near-IID** unlearning and retention evaluation dataset, our method matches WGA in forgetting performance but differs in utility preservation - with relative performance depending on how “utility’’ is defined and which utility domains are considered. Detailed results are shown below.
>
>
> ---
>
> ### Realistic Unlearning Tasks:
>
> As summarized in Tables 1 & 2 below, prior work does not consistently maintain performance across realistic conceptual unlearning tasks:
> - WGA matches CATNIP on MUSE-Book forgetting performance but exhibits notable over-forgetting of general knowledge (MMLU) on WMDP-Cyber and WMDP-Bio tasks.
> - SatImp performs well on WMDP-Cyber but shows catastrophic forgetting on WMDP-Bio and underperforms CATNIP on MMLU within the MUSE-Book setting.
>
>
> In contrast, CATNIP performs well across all three tasks, further supporting the reliability and robustness of our method.
>
>
> **Table1: WMDP Performance Comparison.**
>
> | Methods                     | Bio ↓ | Δf ↓ | MMLU↑ | Δu ↑ | ΔO ↑ | Cyber ↓ | Δf ↓ | MMLU↑ | Δu ↑ | ΔO ↑ |
> |-----------------------------|-------|------|--------|-------|--------|----------|------|--------|-------|--------|
> | **Base model**              | 63.70 | -    | 58.10  | -     | -      | 44.00    | -    | 58.10  | -     | -      |
> | RMU (w/ $D_{retain}$)            | 31.89 | (✓)  | 57.18  | (✓)   | 30.89  | 26.93    | (✓)  | 57.81  | (✓)   | 16.78  |
> | GA + KL (w/ $D_{retain}$)        | 62.77 | (✗)  | 57.29  | (✓)   | 0.12   | 40.36    | (✗)  | 59.82  | (✓)   | 5.36   |
> | NPO + KL (w/ $D_{retain}$)       | 63.16 | (✗)  | 57.67  | (✓)   | 0.11   | 39.61    | (✗)  | 57.11  | (✗)   | 3.30   |
> | FLAT (w/ $D_{ct}$)               | 25.61 | (✓)  | 27.16  | (✗)   | 7.15   | 24.51    | (✓)  | 23.24  | (✗)   | -15.37 |
> | RMU*                        | 25.84 | (✓)  | 25.50  | (✗)   | 5.26   | 24.61  | (✓)  | 25.50  | (✗)   | -13.21 |
> | GA                          | 24.65 | (✓) | 25.25 | (✗)   | 2.60   | 33.77    | (✗)  | 48.79  | (✓)   | 0.92   |
> | NPO                         | 62.69 | (✗)  | 56.88 | (✓) | -0.21 | 36.89    | (✗)  | 55.34 | (✓) | 4.35   |
> | SimNPO                      | 27.10 | (✓)  | 47.37  | (✗)   | 25.87  | 34.22    | (✗)  | 54.25  | (✓)   | 5.93   |
> | WGA                         | 24.59 | (✓)  | 23.31  | (✗)   | 4.30   | 26.07    | (✓)  | 41.30  | (✗)   | 1.13   |
> | SatImp                      | 24.27 | (✓)  | 26.27  | (✗)   | 7.60   | 29.79    | (✓)  | 52.99  | (✓)   | 9.10   |
> | **CATNIP (Ours)**           | 28.36 | (✓) | 51.37 | (✓) |   28.61   | 28.69 | (✓) | 53.01 | (✓) | **10.22** |
>
> **Table2: MUSE-Book Performance Comparison.**
> | Method                       | Know f ↓ (Extended) | Δf ↓ (Extended) | Know f ↓ (MUSE) | Δf ↓ (MUSE) | MMLU ↑ | Δu ↑ | ΔO ↑ |
> |------------------------------|---------------------|------------------|------------------|--------------|--------|-------|-------|
> | **Base model**               | 39.99               | -                | 32.13            | -            | 60.45  | -     | -     |
> | GA + KL (w/ D_r)             | 38.29               | (✗)              | 27.20            | (✗)          | 60.18 | (✓)     | 1.43  |
> | NPO + KL (w/ D_r)            | 33.62               | (✗)              | 28.92            | (✗)          | 59.47  | (✓)     | 5.39  |
> | FLAT (w/ D_ct)               | 5.44                | (✓)              | 6.35             | (✓)          | 50.12  | (✗)   | 24.22 |    |
> | GA                           | 0.00           | (✓)              |     0.00         | (✓)          | 24.87  | (✗)     | -5.61 |
> | NPO                          | 25.21               | (✗)              | 24.18            | (✗)          | 54.79  | (✓)  | 9.12  |
> | SimNPO                       | 6.87                | (✓)              | 6.54             | (✓)          | 51.84  | (✓)     | 24.21 |
> | WGA                          | 2.09                | (✓)              | 3.25             | (✓)          | 40.24  | (✗)     | 27.85 |
> | SatImp                       | 0.00            | (✓)              | 0.00         | (✓)          | 41.84  | (✗)     | 21.38 |
> | **CATNIP (Ours)**            | 2.29            | (✓)              |  2.08        | (✓)          | 52.17  | (✓)     | **29.42** |

---

> ### Author Response · Authors · 2025-11-24
> **Response to Reviewer prd3 (3/4)**
>
> ### Synthetic Unlearning Tasks:
>
> We report five key metrics for this benchamrk:
>
> **Unlearning Performance:** ES forget (exact), ES forget (perturb), and Forget Quality (FQ), where FQ is a $p$-value: $p > 0.05$ indicates the difference between unlearned model and perfectly retained model is not significant, and $p < 0.05$ indicates the difference between unlearned model and perfectly retained model is significant.
>
> **Utility Preservation:**
>
> - MU: Harmonic average across 3 utility metrics spanning 3 domains—synthetic retention data, real author information, and world facts (9 values in total).
> - MU': MU excluding synthetic retention data metrics, which have near-iid distribution with forgetting data.
>
> Note that all methods evaluated use a loss on the retention training data to achieve meaningful MU metrics.
>
> **Results:** As shown in Table 3, regarding **unlearning performance**,  CATNIP achieves comparable or better ES forget performance than WGA as forgetting data volume increases, while WGA demonstrates comparable or better FQ scores than CATNIP.
>
> For **utility perservation**, WGA outperforms CATNIP on MU but underperforms on MU' as forget set size increases. The advantage of  WGA on the MU is partially attributed to the synthetic retention data, suggesting its strength in entry-wise unlearning scenarios while maintaining knowledge on similar retention samples. In contrast, CATNIP better preserves utility on domains semantically distinct from the forgetting dataset.
>
> SatImp exhibits over-forgetting as forgetting data volume increases, with MU/MU' metric declining sharply.
>
> Table 3.1: Comparison between unlearning objectives on TOFU Forget 1% setting using Phi-1.5B. \*indicates the results come from corresponding paper.
> | Method             | ES forget↓ (exact) | ES forget↓ (perturb) | FQ > 0.05? | MU ↑    | MU' ↑   |
> |--------------------|----------------|------------------|------|---------|---------|
> | before unlearning  | 0.5684         | 0.1894           | ✗    | 0.5217  | 0.4616 |
> | WGA                | 0.0079         | 0.0113           | ✓    | 0.5248  | 0.4609 |
> | WGA*               | 0.0344         | 0.0282           | ✓    | 0.5191  | -      |
> | SatImp             | 0.0229         | 0.0206           | ✓    | 0.4422  | 0.4181 |
> | SatImp*            | 0.0464         | -                | ✓    | 0.5248  | -      |
> | CATNIP             | 0.0111         | 0.0195           | ✓    | 0.4922  | 0.4528 |
>
> Table 3.2: Comparison between unlearning objectives on TOFU Forget 5% setting using Phi-1.5B. \*indicates the results come from corresponding paper.
> | Method             | ES forget↓ (exact) | ES forget↓ (perturb) | FQ >0.05? | MU ↑    | MU' ↑   |
> |--------------------|----------------|------------------|------|---------|---------|
> | before unlearning  | 0.6114         | 0.1814           | ✗    | 0.5217  | 0.4616 |
> | WGA                | 0.0232         | 0.0227           | ✓    | 0.5166  | 0.4601 |
> | WGA*               | 0.0179         | 0.0199           | ✗    | 0.5108  | -      |
> | SatImp             | 0.0000         | 0.0000           | ✗    | 0.2095  | 0.2241 |
> | SatImp*            | 0.0427         | -                | ✗    | 0.5214  | -      |
> | CATNIP             | 0.0172         | 0.0143           | ✗    | 0.4173  | 0.4404 |
>
> Table 3.3: Comparison between unlearning objectives on TOFU Forget 10% setting using Phi-1.5B. \*indicates the results come from corresponding paper.
> | Method             | ES forget↓ (exact) | ES forget↓ (perturb) | FQ >0.05?  | MU ↑    | MU' ↑   |
> |--------------------|----------------|------------------|------|---------|---------|
> | before unlearning  | 0.5617         | 0.1960           | ✗    | 0.5217  | 0.4616 |
> | WGA                | 0.0328         | 0.0301           | ✗    | 0.5157  | 0.4695 |
> | WGA*               | 0.0000         | 0.0000           | ✗    | 0.5183  | -      |
> | SatImp             | 0.0000         | 0.0000           | ✗    | 0.0812  | 0.0638 |
> | SatImp*            | 0.0407         | -                | ✗    | 0.5107  | -      |
> | CATNIP             | 0.0261         | 0.0272           | ✗    | 0.4842  | 0.4849 |
>
> Please note that:
> *1. For WGA [1] and SatImp [2], we report both their originally published results (denoted as WGA$^\*$, SatImp$^\*$) and our reproductions using their source code*.
> 2. SatImp's reproduced performance differs substantially from [2], likely due to undisclosed hyperparameter configurations in their implementation.
>
> ---
>
> We are running an additional experiment on larger-scale LLMs to further validate the generality of our findings. Due to computational cost, this experiment is still in progress. Importantly, all existing evidence presented in the rebuttal has supported our conclusions, and the preliminary results we have obtained so far are consistent with the reported trends.

---

> ### Author Response · Authors · 2025-11-24
> **Response to Reviewer prd3 (4/4)**
>
> Once again, we sincerely appreciate you for the valuable feedback and constructive questions. We look forward to any further feedback you may have.
>
> Sincerely,
> Authors
>
> ---
>
> [1] Rethinking LLM Unlearning Objectives: A Gradient Perspective and Go Beyond. ICLR2025
>
> [2] Exploring Criteria of Loss Reweighting to Enhance LLM Unlearning. ICML2025
>
> [3] RULE: Reinforcement UnLEarning Achieves Forget–retain Pareto Optimality. NeurIPS 2025.

---

> > ### Comment · Reviewer_prd3 · 2025-11-25
> >
> > ## Post-rebuttal Comments
> >
> > I appreciate the authors’ response and the additional clarifications. It is encouraging to see that the paper now acknowledges and discusses prior works that should have been considered before submission.
> >
> > As clarified in the rebuttal, the proposed method differs from [1] and [2] primarily at a *conceptual* level. Summarizing the weighting formulations for clarity:
> > - **WGA:**       $p^{\beta}$
> > - **SatImp:**       $p^{\beta_1}(1 - p)^{\beta_2}$
> > - **This paper:**  $\frac{p^{\beta}}{p^{\beta} + (1 - p)^{\beta}}$
> >
> > This comparison makes the extent of the paper’s contribution more transparent. **Beyond a well-crafted narrative, the substantive novelty lies mainly in this particular weighting form.** While this might represent a substantial improvement compared to earlier *sample-wise* methods, **its difference from [1] and [2] is marginal**. The proposed formulation can thus be viewed as a smooth variant rather than a fundamentally new idea. Consequently, the method’s novelty appears limited, and its performance is only **competitive** rather than **significantly outperforming** existing approaches.
> >
> > ---
> >
> > ### On the TOFU Dataset
> >
> > Regarding the second point on the **TOFU dataset**, I find the authors’ rebuttal unconvincing. The argument that TOFU is “not realistic” because it is *synthesized* rather than derived from real-world data is weak. The practicality of TOFU lies not in whether its content is human-generated, but in the **scenario it represents** — a model that encapsulates broad, domain-level knowledge, similar to real-world LLMs fine-tuned for specific applications. In that sense, TOFU is a *highly relevant and realistic* benchmark for evaluating unlearning behavior in models with complex knowledge entanglement.
> >
> > If the authors had instead argued that TOFU’s **scale** or **data coverage** is limited, the point would have been more acceptable. However, dismissing TOFU’s realism entirely feels like an excuse for not considering a more comprehensive evaluation.
> >
> > ---
> >
> > ### Overall Assessment
> >
> > After considering the authors’ rebuttal and the detailed points discussed above, I believe this paper represents a rather **immature or incomplete stage of research**. It appears to be a concurrent effort that may have been initiated before the publication of WGA and SatImp, leading to **experimental setups and design choices that now seem underdeveloped**. The current submission appears to have been **resubmitted with only minor revisions**, without sufficient attention to recent and highly relevant works that should have been included for fair comparison.
> >
> > Regarding the authors’ comment in the rebuttal that *“SatImp’s hyperparameters are not clearly defined”*, this is inaccurate. Both **WGA** and **SatImp** are implemented and discussed within the **open-unlearning framework** [3], where their hyperparameter configurations are explicitly documented. I do not believe the authors intentionally avoided comparison with [1] or [2]; rather, this reflects a **lack of up-to-date understanding of recent advances in the field**, which in turn affects their ability to accurately evaluate the novelty and contribution of their own work.
> >
> > As the field of LLM unlearning progresses rapidly, most of the contributions and performance improvements presented in this paper have already been matched or surpassed by more recent studies. Consequently, both the **novelty** and the **relative performance advantage** of this work have diminished over time.
> >
> > If this paper had been submitted to **ICLR 2025**, it could have been considered a promising and timely contribution. Unfortunately, as of **ICLR 2026**, the overall contribution no longer meets the standard expected of a top-tier machine learning venue. I would therefore recommend the authors consider submitting this work to **ACL** or **EMNLP**, where the topic and level of contribution may be a better fit.
> >
> > ---
> >
> > **Therefore, my overall assessment remains unchanged (I would prefer a score of 3, but ICLR 2026 does not include this option). While the paper is clearly written and addresses a meaningful topic, its conceptual novelty and empirical contribution do not reach the threshold required for acceptance at ICLR.**
> >
> >
> > [1] Rethinking LLM Unlearning Objectives: A Gradient Perspective and Go Beyond. ICLR2025
> >
> > [2] Exploring Criteria of Loss Reweighting to Enhance LLM Unlearning. ICML2025
> >
> > [3] https://github.com/locuslab/open-unlearning/tree/main

---

> ### Author Response · Authors · 2025-11-26
> **Response to Post-rebuttal Comments (1/2)**
>
> We thank reviewer rpd3 for the detailed feedback and for the continued engagement. We would like to clarify several key points below.
>
>
> ### 1. Theoretical Contribution
>
>
> 1. We appreciate the reviewer for listing the final gradient expression of different works, which is one axis of comparison. However, our contribution is not limited to proposing a new weighted gradient form. The key distinction lies in the implications behind these gradient weights. WGA [1] directly crafts an importance weight before the log of policy gradient $\log \pi_\theta$, named as *G-effect*, without theoretically interpreting **what machine learning objectives are really being optimized** when different *G-effect* weights are applied. SatImp [2] extended WGA [1] by proposing two heuristic concepts: saturation $\pi_\theta^{\beta_1}$ and importance $(1 - \pi_\theta)^{\beta_2}$, and combined them as the new gradient weights different from WGA.
>
>
>
>
>
> 2. Instead of taking bottom-up approaches, our method is top-down, derived from a policy-ranking preference objective, which clearly **interprets** the weighted gradient function that is derived. This formulation can incorporate prior heuristic concepts into one unified framework, with a chosen reverse policy as the reference model that dynamically reflects unlearning importance and saturation. Case studies (Fig. 2 in the main paper and Table 10 - 11 in the Appendix) have shown how this methodology leads to faster and calibrated unlearning of important tokens related to unlearning domain knowledge.
>
>
>
> ### 2. Empirical Advantages:
>
> Experiments have shown that our proposed method can outperform prior work across key benchmarks and metrics. Especially:
>
> #### WMDP benchmark:
>
>   * MUSE-Bio (**ΔO ↑**): **28.61** (Ours)  > 7.60 (SatImp) >  4.3 (WGA)
>   * MUSE-Cyber (**ΔO ↑**): **10.22** (Ours) > 9.10 (SatImp) > 1.13 (WGA)
>
>
> #### MUSE-Book benchmark:
>
> * **ΔO ↑**: **29.42** (Ours) > 27.85 (WGA) > 14.34 (SatImp)
>
>
> #### TOFU benchmark:
>
>
> - $10\%$ **ES forget exact (↓)**: **0.0261** (Ours) < 0.0658 (SatImp)  < 0.0328 (WGA)
>
> - $10\%$ **MU' ↑**:  **0.4849** (Ours) > 0.4695 (WGA) > 0.4579 (SatImp)
>
>
>
> **Reproducible issue of satimp:**
>
>
> Thank you very much for pointing out OpenUnlearning [3] and the potential hyperparameter solution. We would like to clarify the following points:
>
> 1. Although OpenUnlearning [3] mentions WGA [1] and SatImp [2], it does not report experimental results for these methods. Our original implementation of SatImp followed the official codebase released by [2].
> 2. We acknowledge that the OpenUnlearning community provides an alternative hyperparameter configuration for SatImp, which we have faithfully reproduced. Specifically, in our original setup, denoted as SatImp, we followed the default hyperparameters from [2], setting both $\beta_1$ and $\beta_2$ to 0.3. The configuration provided by the OpenUnlearning community adopts $\beta_1 = 5.0$ and $\beta_2 = 1.0$. In both settings, the forgetting loss to retention loss ratio is set to 10. The detailed results are updated in Table 4.
>
>
>
>
>
> ### 3. Preferences of Unlearning Benchmarks:
>
> We appreciate the reviewer's perspective, but we maintain our position based on our research design principle, not convenience. We clarify our reasoning below:
>
> 1. We target the unlearning scenario where:
>
>    **i)** The forgotten knowledge should be realistic and exist in the base LLM. The forgetting data of TOFU are **synthetic QA samples of fake authors.** (Section 2.2.1 in TOFU paper [4]), and the knowledge **does not exist naturally in LLMs**, and in order to conduct experiments on the TOFU benchmark, an enhanced model with synthetic forget data and retention data needs to be trained first as the base model (Section 4 in [4]). The synthetic setting is one of the reasons that have motivated subsequent unlearning benchmarks, which explicitly highlight and propose realistic knowledge data naturally occurring in LLMs, such as WMDP (Section 2 in [5]) and MUSE (Section 1, Table 1 in [6]);
>
>    **ii)** The retained knowledge is semantically separable from the forgotten one. The Model Utility (MU) metric in TOFU contains three types of retention data: a) Synthetic retention data that is iid distributed with forgetting data (Section 2.1 in [4]), since the same synthetic dataset was split into the forget and retention subsets; b) Real author information; and c) World facts. We used the latter two for retention metric (MU') evaluation as they are real knowledge that semantically differ from fake author information, and usually naturally exist in LLMs.

---

> ### Author Response · Authors · 2025-11-26
> **Response to Post-rebuttal Comments (2/2)**
>
> ---
>
> **Table 4: Performance Comparison on TOFU benchmark (updated).**
>
> Table 4.1: Comparison between unlearning objectives on TOFU Forget 1% setting using Phi-1.5B. \*indicates the results come from corresponding paper.
>
> | Method            | ES forget↓ (exact) | ES forget↓ (perturb) | FQ > 0.05? | MU ↑   | MU' ↑  |
> | ----------------- | ------------------ | -------------------- | ---------- | ------ | ------ |
> | before unlearning | 0.5684             | 0.1894               | ✗          | 0.5217 | 0.4616 |
> | WGA               | 0.0079             | 0.0113               | ✓          | 0.5248 | 0.4609 |
> | WGA*              | 0.0344             | 0.0282               | ✓          | 0.5191 | -      |
> | SatImp            | 0.0816             | 0.2006               | ✗          | 0.5244 | 0.4685 |
> | SatImp*           | 0.0464             | -                    | ✓          | 0.5248 | -      |
> | CATNIP            | 0.0111             | 0.0195               | ✓          | 0.4922 | 0.4528 |
>
> Table 4.2: Comparison between unlearning objectives on TOFU Forget 5% setting using Phi-1.5B. \*indicates the results come from corresponding paper.
>
> | Method            | ES forget↓ (exact) | ES forget↓ (perturb) | FQ >0.05? | MU ↑   | MU' ↑  |
> | ----------------- | ------------------ | -------------------- | --------- | ------ | ------ |
> | before unlearning | 0.6114             | 0.1814               | ✗         | 0.5217 | 0.4616 |
> | WGA               | 0.0232             | 0.0227               | ✓         | 0.5166 | 0.4601 |
> | WGA*              | 0.0179             | 0.0199               | ✗         | 0.5108 | -      |
> | SatImp            | 0.1990             | 0.0686               | ✗         | 0.5092 | 0.4579 |
> | SatImp*           | 0.0427             | -                    | ✗         | 0.5214 | -      |
> | CATNIP            | 0.0172             | 0.0143               | ✗         | 0.4173 | 0.4404 |
>
>
>
> Table 4.3: Comparison between unlearning objectives on TOFU Forget 10% setting using Phi-1.5B. \*indicates the results come from corresponding paper.
>
> | Method            | ES forget↓ (exact) | ES forget↓ (perturb) | FQ >0.05? | MU ↑   | MU' ↑  |
> | ----------------- | ------------------ | -------------------- | --------- | ------ | ------ |
> | before unlearning | 0.5617             | 0.1960               | ✗         | 0.5217 | 0.4616 |
> | WGA               | 0.0328             | 0.0301               | ✗         | 0.5157 | 0.4695 |
> | WGA*              | 0.0000             | 0.0000               | ✗         | 0.5183 | -      |
> | SatImp            | 0.0658             | 0.0660               | ✗         | 0.5044 | 0.4579 |
> | SatImp*           | 0.0407             | -                    | ✗         | 0.5107 | -      |
> | CATNIP            | 0.0261             | 0.0272               | ✗         | 0.4842 | 0.4849 |
>
> *1. For WGA [1] and SatImp [2], we report both their originally published results (denoted as WGA\* and SatImp\*) and our reproductions. Specifically, SatImp was reproduced using hyper parameter setting from paper [3] as suggested by reviewer rpd3.*
>
>
> ---
> Thanks again for your participation in the discussion and your valuable feedback. I hope our response addresses your concern. We look forward to any further feedback you may have.
>
> Sincerely,\
> Authors
>
> [4] TOFU: A Task of Fictitious Unlearning for LLMs. COLM2024\
> [5] The WMDP Benchmark: Measuring and Reducing Malicious Use with Unlearning. ICML2024\
> [6] MUSE: Machine Unlearning Six-Way Evaluation for Language Models. ICLR2025

---

> > ### Comment · Reviewer_prd3 · 2025-11-26
> >
> > ### Thank you for the authors’ response. Unfortunately, the rebuttal reads more like an attempt to retrofit explanations rather than addressing the core issues raised. To avoid further misunderstanding, I restate my concerns below in a more direct and explicit manner.
> >
> > -----
> >
> > ### Theoretical Contribution
> >
> > As mentioned earlier, the claimed novelty essentially reduces to adopting a weighting strategy that differs only superficially from WGA and SatImp. Throughout the rebuttal, the authors repeatedly emphasize the motivation and supposed “theoretical” differences. However, this does not address the core issue: **the methodological substance is minimal**.
> >
> > To be very clear—this is ICLR, not an NLP venue where proposing a new CoT prompt template might suffice as a main-track contribution [1]. At ICLR, reviewers focus on **actual algorithmic** or **theoretical innovation**, not on narrative framing. What matters is the underlying mechanism, not the story wrapped around it.
> >
> > Fundamentally, the idea boils down to a simple optimization objective based on $\(1 - p\)$, which is conceptually equivalent to the $\(1 - p\)$ component in SatImp. Replacing explicit $\(1 - p\)$ with a model’s reverse-output probability does not constitute meaningful novelty. This is a cosmetic modification, not a conceptual advance.
> >
> > My earlier point remains: in the current landscape of LLM unlearning research, the era where a single reweighting trick—showing marginal gains on a few datasets—could qualify as a top-tier machine learning contribution has passed. The bar has moved significantly, and the proposed method does not meet the level of conceptual depth or theoretical insight expected at ICLR.
> >
> > -------
> >
> > ### Performance
> >
> >
> > Based on my inspection of prior results, the WMDP benchmark is typically used in such a way that the Bio and Cyber subsets are optimized **jointly**.
> > In this work, however, authors separate them into two distinct sub-tasks for independent evaluation and analysis.
> >
> > Besides,
> >
> > 1. results **without** retain regularization.
> > 2. results **with** retain regularization.
> >
> > These two variants differ substantially in their MMLU outcomes. If the authors’ method aims to compete fairly with existing baselines, then the comparison must be made against **both** variants.
> >
> > Without this clarification, the empirical comparison is **ambiguous and potentially misleading**. If the proposed method is compared against only one version, the conclusions drawn about “outperforming” or being “competitive” may not hold under the other, equally standard, evaluation setting.
> >
> >
> > ### SatImp hyper-parameter
> >
> > The authors’ rebuttal regarding the SatImp hyperparameters is deeply concerning. Their first response claimed that the hyperparameters “could not be found,” and the second response stated that the values should be “$\beta_1 = 0.3, \beta_2 = 0.3$.” To verify this, I revisited the SatImp paper. On page 7, in the **Experiments** paragraph, the hyperparameter choice is stated very clearly:
> >
> > > “ The hyper-parameter for SatImp is set to $\beta_1 = 5$,$\beta_2 = 1$”
> >
> > **Given how prominently this information appears, it is difficult to understand why the authors first claimed the hyperparameters were missing, and later cited completely different values. These inconsistent statements seriously call into question whether the authors have carefully read the SatImp paper at all.**
> >
> > Additionally, I am puzzled by the choice of setting the forget-to-retain ratio to 10. It is well understood that there is a trade-off between forgetting and retention. Why is the ratio 10 instead of, for example, 0.1 or 1? The open-unlearning framework explicitly discusses the importance of balanced ratios (In open-unlearning, this forget/retain ratio is set to 0.1). A ratio of 10 strongly biases the model toward over-unlearning, which raises concerns about the **fairness** and **interpretability** of the reported results. **Without justification, such a configuration appears arbitrary and potentially misleading.**
> >
> > For completeness, I also reproduced WGA and SatImp using the official code in [2]. However, the performance on the TOFU benchmark I obtained is as follows:
> >
> > Phi-1.5
> >
> > Forget 5%:
> >
> > WGA: ES_Forget=0.0306 MU=0.5237
> >
> > Satimp: ES_Forget=0.0187 MU=0.5295
> >
> > Forget 10%:
> >
> > WGA: ES_Forget=0.0000 MU=0.5305
> >
> > Satimp: ES_Forget=0.0000 MU=0.5157
> >
> > [1] LLM Unlearning Without an Expert Curated Dataset. arXiv:2508.06595
> >
> > [2] Exploring Criteria of Loss Reweighting to Enhance LLM Unlearning. ICML2025

---

> > > ### Comment · Reviewer_prd3 · 2025-11-26
> > >
> > > ### Preferences of Unlearning Benchmarks
> > >
> > > The rebuttal reiterates the authors’ original position without engaging with the substantive concerns raised. The statement “we maintain our position based on our research design principle” is particularly aggressive and problematic. Such phrasing suggests that the authors’ position is to be accepted as inherently correct, rather than justified through empirical evidence or theoretical reasoning. Scientific arguments require support, not assertion.
> > >
> > > The authors’ justification for excluding TOFU—namely that TOFU contains synthesized knowledge that “does not naturally exist in base LLMs”—is based on a fundamentally flawed premise. The assumption that unlearning should only be evaluated on “naturally existing” LLM knowledge does not cover all of real-world scenarios.
> > >
> > > Consider a realistic scenario:
> > > A base LLM model $\theta_0$ performs poorly in a specialized domain (e.g., medical), so a practitioner fine-tunes it to obtain $\theta_1$. The knowledge in $\theta_1$ did not exist in $\theta_0$, yet it is unquestionably real, practical, and relevant. If the authors’ criterion were applied consistently, such specialized fine-tuning knowledge would also be considered “invalid” for unlearning evaluation simply because it did not originate in the base model.
> > >
> > > This "position that based on our research design principle" is internally inconsistent. From the model’s perspective, both TOFU knowledge and domain-specific fine-tuned knowledge are initially unknown, and after fine-tuning, both become integrated model knowledge. The distinction the authors draw between “synthetic” and “real-world” is therefore irrelevant at the model level and does not serve as a meaningful basis for excluding TOFU.
> > >
> > > More importantly, TOFU represents a plausible and widely acknowledged real-world scenario: an LLM possessing structured, domain-level knowledge that must be selectively removed. Many prior works have used synthesized or constructed datasets for precisely this purpose. Rejecting TOFU on semantic grounds (“facts” vs “non-facts”) implies that all previous benchmark-driven unlearning research is invalid, which is clearly untenable.
> > >
> > > In short, the rebuttal does not provide a coherent rationale for excluding TOFU; instead, it relies on a semantic distinction that does not align with how LLMs acquire, store, or utilize knowledge in practical systems. The position is neither empirically nor conceptually well supported.
> > >
> > > --------
> > > Up to this point, I have addressed all components of the authors’ rebuttal with clear and detailed explanations. Unfortunately, despite the polite tone of the response, I see **no substantive improvement** in the scientific clarity or rigor of the work. I would like to highlight the following concerns to the Area Chair:
> > >
> > > 1. **Why were WGA and SatImp—two highly relevant token-wise baselines—not cited or compared against in the original submission?**
> > >    Only after receiving the reviews did the authors shift their narrative toward “conceptual novelty,” rather than acknowledging the absence of genuinely new findings that would justify a new method.
> > >
> > > 2. **Why were the SatImp hyperparameters misunderstood in two consecutive rebuttals**, even though the correct values are stated explicitly and prominently in the original paper?
> > >    This raises serious doubts about whether the authors carefully read the related work they were asked to compare against.
> > >
> > > 3. **Why do the authors’ reproduced results conveniently outperform WGA and SatImp by a small margin, especially under an evidently over-unlearning configuration?**
> > >    Under such settings, both WGA and SatImp still retain residual knowledge, but the proposed method appears to surpass them precisely under this biased setup. Furthermore, the authors’ reported numbers are noticeably lower than what I obtain when reproducing the results using the official implementation, creating additional concerns about reproducibility and fairness.
> > >
> > > 4. **Why do the authors appear unwilling to engage with constructive feedback?**
> > >    The rebuttal demonstrates a rigid adherence to their own assumptions about benchmark design, without acknowledging valid criticisms or engaging in scientific dialogue. The sentence *“We appreciate the reviewer's perspective, but we maintain our position based on our research design principle, not convenience.”* reads less like an explanation and more like a dismissal of reviewer concerns.
> > >
> > > Given all of the above, I must be explicit for the Area Chair:
> > > **The authors’ rebuttal does not address the raised concerns; instead, it further undermines my confidence in the maturity, rigor, and reliability of this work.**

---

> > > > ### Author Response · Authors · 2025-11-27
> > > > **Response to prd3 round3**
> > > >
> > > > We apologize if our previous responses were perceived as dismissive. That was never our intention. We deeply value the peer review process and the opportunity for engaged discussion. We should have been clearer in distinguishing between explaining our experiment choices and remaining open to alternative perspectives.
> > > >
> > > > #### 1. Revisions Made in Response to Reviewer Feedback
> > > >  We have already made substantial changes to our manuscript based on reviewer feedback:
> > > >
> > > > - Added missing baselines and comparisons.
> > > > - Expanded experimental evaluation (including additional TOFU experiments in A.9).
> > > > - Clarified hyperparameter choices and their sources.
> > > > - Improved related work discussion.
> > > >
> > > >
> > > >
> > > > #### 2. On Theoretical Contribution and Novelty
> > > > Regarding the significance of theoretical contributions, we respect your assessment while hoping to provide sufficient justification for ours. Ultimately, we defer judgments of novelty to the community.
> > > >
> > > > * Regarding the comments of reverse **reference policy** $(1 - \pi_\theta)$ in our formulation *vs.* the **gradient weight** $(1 - \pi_\theta)$ in SatImp: "*Fundamentally, the idea boils down to a simple optimization objective based on , which is conceptually equivalent to the component in SatImp. Replacing explicit with a model’s reverse-output probability does not constitute meaningful novelty.*"
> > > >
> > > > **Response**: These two are **not directly comparable**. The derivation process from Equation 8 to Equation 9 in our paper proved that using this reference policy does not recover the gradient weights designed in SatImp or WGA.
> > > >
> > > > #### 3. Regarding TOFU and SatImp Experiments
> > > >
> > > > **SatImp hyperparameter configurations**: We acknowledge the confusion surrounding hyperparameter choices and clarify that it arose from **inconsistencies across multiple authoritative sources**:
> > > >
> > > >
> > > >
> > > >   - Official SatImp paper [2]: "The hyper-parameter for SatImp is set at $\beta_1=5$, $\beta_2=1$ "
> > > >   - Official SatImp code [2a]:  $\beta_1=\beta_2=0.3$
> > > >   - OpenUnlearning benchmark [3]: $\beta_1=5$, $\beta_2=1$
> > > >
> > > > Timeline:
> > > >   - Response round \#1, we used hyper-parameter settings provided in [2a], with $\beta_1=\beta_2$ = 0.3
> > > >   Response round \#2: As reviewer suggested, we used  hyper-parameter settings provided in OpenUnlearning benchmark [3a] - which is consistent with the hyper description of [2] with $\beta_1=5$ and  $\beta_2=1$.
> > > >
> > > > **Forget-to-retain ratio of SatImp**: For the forget-to-retain ratio of 10, we followed the experimental setup described in [3a] and the implemention of [2a], which set 1 as the weight of forgetting loss and set 0.1 as the weight of retention loss.
> > > >
> > > > **SatImp performance inconsistency**:  We note that independently reproduced SatImp results can vary non-trivially, from those reported in Table 2 of the original paper \[2\], to the one reproduced by reviewer prd3, depending on implementation and evaluation details.
> > > >
> > > > Phi-1.5\
> > > > Forget 5%:\
> > > > WGA (reviewer prd3): ES_Forget=0.0306 MU=0.5237
> > > > WGA \[1\]: ES_Forget=0.0179 MU=0.5108
> > > > Satimp (reviewer prd3): ES_Forget=0.0187 MU=0.5295
> > > > Satimp \[2\]: ES_Forget=0.0427 MU=0.5214
> > > >
> > > > Forget 10%:\
> > > > WGA (reviewer prd3): ES_Forget=0.0000 MU=0.5305
> > > > WGA \[1\]: ES_Forget=0.0000 MU=0.5183
> > > > Satimp(reviewer prd3) : ES_Forget=0.0000 MU=0.5157
> > > > Satimp \[2\]: ES_Forget=0.0407 MU=0.5107
> > > >
> > > > To support a fair assessment, we have clarified our reproduction protocol and experimental settings in the revised manuscript.
> > > >
> > > >
> > > > \[2a\]: https://github.com/tmlr-group/SatImp/tree/main
> > > >
> > > > \[3a\]: https://github.com/locuslab/open-unlearning/tree/main/community/methods/SatImp
> > > >
> > > >
> > > > ---
> > > > We are happy to address any additional questions or comments from the reviewers.
> > > >
> > > > Sincerely,\
> > > > Authors

---

> > > > > ### Comment · Reviewer_prd3 · 2025-11-27
> > > > >
> > > > > ## Thank you for the additional response. I want to emphasize that none of my comments were personal; they are strictly about the scientific aspects of the work. We are both presenting our understanding of the LLM unlearning, and it is natural that strong opinions arise in such discussions. I am glad that certain factual points have been clarified, and it is also reasonable that some disagreements remain.
> > > > >
> > > > > -------
> > > > >
> > > > > ### About modifications in rebuttal
> > > > >
> > > > > No one is denying that you have made changes, but whether these changes substantially improve the contribution of the paper is debatable. In fact, it is precisely the newly added clarifications and comparisons that reveal the actual contribution of the work at this stage. If we consider only the original submission, it creates the impression that this is the first paper to apply token-wise weighting for unlearning, with performance significantly surpassing prior methods. After the revisions, however, it becomes clear that—inevitably—the perceived novelty at the methodological level and the magnitude of empirical improvements are both considerably diminished.
> > > > >
> > > > > ---------
> > > > >
> > > > > ### About Novelty
> > > > >
> > > > > I fully agree with one sentence from the authors: **“we defer judgments of novelty to the community.”** This is indeed an area where we cannot reach consensus, because our starting points differ. Just as the authors are confident in their position, I am equally confident in mine. In my earlier reviews, all the related works I referenced are already published. If the difference is claimed to lie in **gradient-based reverse reweighting** versus **reference-model-based reverse reweighting**, then I can state clearly that **this paper is not the first** to do so. The essential idea already appears in [1], which also uses reference-model-based token-wise reverse reweighting. I know the authors will likely argue that their formulation is still different, so if they wish to respond, they do not need to address this specific point—we can indeed leave this judgment to the community.
> > > > >
> > > > > [1] Distribution Preference Optimization: A Fine-grained Perspective for LLM Unlearning  ArXiv (NeurIPS2025 Submission)
> > > > >
> > > > > -------
> > > > >
> > > > > ### About dataset Choice
> > > > >
> > > > > What I want to emphasize further is the broader context of LLM unlearning as a field. It is evident that this work, like several earlier foundational studies, proposes a competitive reweighting function evaluated on the standard benchmarks (old setting, new methods), achieving competitive performance (and I will elaborate on this below). However, it is also clear that since ICML 2025, the community’s focus has shifted beyond basic performance toward more practical unlearning challenges [2][3] (new setting, new methods). Contemporary works [4] of the same type as this paper (old setting, new method) typically evaluate on all 3 major unlearning benchmarks. Practical works [5][6] (new setting, new methods) choose to use 2 or 3 benchmarks.
> > > > >
> > > > > [2] Towards llm unlearning resilient to relearning attacks: A sharpness-aware minimization perspective and beyond. ICML2025 Workshop
> > > > >
> > > > > [3] Unlearning Isn't Invisible: Detecting Unlearning Traces in LLMs from Model Outputs. ICML2025
> > > > >
> > > > > [4] LLM Unlearning with LLM Beliefs. Arxiv
> > > > >
> > > > > [5] Leak@k: Unlearning Does Not Make LLMs Forget Under Probabilistic Decoding. Arxiv
> > > > >
> > > > > [6] Forgetting to Forget: Attention Sink as A Gateway for Backdooring LLM Unlearning. Arxiv
> > > > >
> > > > > In contrast, the authors’ reluctance toward using TOFU—illustrated by the reasoning they provided, the disagreements in our discussions, and the eventual decision to run TOFU experiments only after repeated prompting—indicates both **a hesitation to engage with this benchmark** and **a lack of familiarity with the most up-to-date directions in unlearning research**. This is reflected in the shifting justifications for excluding TOFU, and in the final response where the authors stated that the matter was simply a difference in opinion, rather than acknowledging that the initial dismissal was not well considered.

---

> > > > > > ### Comment · Reviewer_prd3 · 2025-11-27
> > > > > >
> > > > > > ### About Performance
> > > > > >
> > > > > > Thank you for candidly acknowledging that **you did not consult the original papers when reproducing other methods’ results**; in the end, it is good that we were able to reconcile the inconsistencies in performance. Regarding the ratio issue, this actually relates to the next point I would like to raise.
> > > > > >
> > > > > > The change in the performance claims is something the authors now seem unwilling to acknowledge. As I noted in my second-round response: in the original submission, the method appears to be state-of-the-art when compared only against older sample-wise methods. However, after including comparisons with WGA and SatImp, it becomes clear that **the proposed method is at best competitive, not superior**. We all understand that authors choose the hyperparameters that give the best results for their own method, whereas WGA and SatImp may not have been tuned with the same level of attention. Even if we assume optimal tuning for all methods, the performance improvements of the proposed approach are significantly reduced, lowering the overall contribution of the method from a performance perspective.
> > > > > >
> > > > > > -------
> > > > > >
> > > > > > ### Final Words
> > > > > >
> > > > > > After several rounds of discussion, I believe we have reached some agreements as well as some continued disagreements. I want to restate my central point: **prior to the revisions, the contribution of this paper appeared much stronger than it actually was.** Due to a lack of awareness of the most recent advances in the field, several important existing works were overlooked. During the rebuttal phase, the authors did provide some brief comparisons and discussions, which is positive, but key disagreements remain.
> > > > > >
> > > > > > More importantly, after incorporating the omitted related work and reassessing both the theoretical and empirical contributions, it becomes clear that **the novelty of the proposed method and the extent of its performance improvements are both significantly reduced compared to the original submission.** These reduced contributions fall below my expectations for an ICLR paper. As I stated after the first rebuttal, I believe the work could be publishable at an NLP venue, but it does not meet the standard required for ICLR.
> > > > > >
> > > > > > If my wording or tone made the authors uncomfortable or gave the impression that I was intentionally making things difficult, I am sorry you feel that way. From the authors’ final response, it is clear that they found it difficult to accept that I have raised many concerns without increasing the score, and some of the phrasing reflects this sentiment.
> > > > > >
> > > > > > **To show my respect for the authors’ substantial effort on this work, I will raise my score to 4.** This is my final decision and will not be changed further. **I would like the Area Chair to note that this score increase reflects only my recognition of the authors’ hard work; our disagreements remain significant, and I still firmly believe that the overall contribution of the paper does not meet the standard for ICLR.**
> > > > > >
> > > > > > Thank you to the authors for their efforts. And, more than that the authors said, we defer all of judgments to the community.

---

### Official Review · Reviewer_JFzo · 2025-10-31

**Soundness:** 3
**Presentation:** 2
**Contribution:** 2
**Rating:** 4
**Confidence:** 3

**Summary:**

This paper proposes CaTNiP, a new method for LLM unlearning, i.e., selectively removing specific undesirable knowledge (e.g., copyrighted or hazardous information) from pretrained large language models.

Extensive experiments on WMDP (hazardous knowledge) and MUSE (copyrighted text) benchmarks show that CaTNiP achieves superior unlearning–retention trade-offs compared to GA, NPO, SimNPO, FLAT, and RMU, even without retention or contrastive data.

**Strengths:**

The paper contributes a novel perspective on LLM unlearning, grounding it in policy-level preference alignment rather than response-level contrastive optimization.

The tokenized unlearning formulation is an insightful response to a real practical issue—sequence-length bias—that plagues many recent alignment and unlearning methods.

The paper provides clear mathematical derivations, including how token-level calibration induces rescaled gradients (Eq. 9) and the monotonic weighting behavior

Experiments： multiple datasets (WMDP–Bio, WMDP–Cyber, MUSE–Books), strong baselines (GA, NPO, SimNPO, FLAT, RMU), and detailed ablations (CATNIPref, CATNIP without tokenization).

**Weaknesses:**

Limited theoretical formulation:
While intuitive, defining the reference policy lacks theoretical justification from probabilistic or game-theoretic perspectives. The authors could better formalize why this choice leads to stable optimization and avoids degeneracy.

Absence of comparisons with more recent LLM editing approaches:
The study focuses on unlearning baselines (GA, NPO, RMU, FLAT) but omits recent localized editing and LoRA-based selective forgetting methods.

Evaluation limited to mid-size models (≤7B).
All experiments are conducted on Llama-3B and Zephyr-7B. It remains unclear whether the calibration dynamics scale to larger instruction-tuned models (e.g., 13B, 70B).

**Questions:**

scalability:  How does CaTNiP perform when applied to larger models (e.g., Llama-13B or Mistral-7B)?

Does the adaptive reference introduce additional computational overhead or memory cost compared to standard NPO?

robustness and composability:  Can CaTNiP be composed sequentially for multiple unlearning tasks (e.g., forgetting multiple domains)?

interpretability:  Could the authors provide visualizations of token-level unlearning beyond the single case study (Fig. 2)?

---

> ### Author Response · Authors · 2025-11-24
> **Response to Reviewer JFzo**
>
> Dear reviewer JFzo,
>
> Thank you for your valuable and thoughtful review. We are encouraged by your recognition of our novel unlearning formulation, theoretical insights, and comprehensive experimental evaluation. Please find below our responses to address your concerns and questions.
>
> ---
>
> **Weakness**: Limited theoretical formulation: While intuitive, defining the reference policy lacks theoretical justification. The authors could better formalize why this choice leads to stable optimization and avoids degeneracy.
>
> **Response**: Thank you for raising this important question. Our choice of $1-\pi_\theta$ as the reference model offers the following  advantages in stablizing optimization:
>
> **(1)** It effectively forms  a *counterfactual*  guidance from the start. In the initial unlearning stage, the target policy $\pi_\theta$ is prone to generating forgetting knowledge. Using $1-\pi_\theta$ is more likely to refuse to generate such content,  providing a strong initial guidance. In contrast, $\pi_\text{ref} = \pi_\theta$ serves as a weaker reference when training starts, as it mirrors rather than counters the models' undesirable behavior.
>
> **(2)**  Our choice of reference model also smoothes the confidence weight and stabilizes training (In our experiments, we always used $\beta >=1$):
>
> - As shown in Figure 1 of the paper, when $\beta >= 1$, the  weight $w_i(\beta, \pi)$ is bounded. Especially, when $\beta>1$, the weight function gets a smooth curve when the current policy $\pi_\theta$ is over-confident in generating or refusing  a token( *i.e.*  $\pi_\theta \to0$ or $\pi_\theta \to 1$).
> - When $\pi_\theta$ reaches maximum uncertainty ( $\pi_\theta \to 0.5$), the weight derivative reaches the highest slope, which can appropriately amplify the learning signal.
>
> ---
>
> **Question**: Does the adaptive reference introduce additional computational overhead or memory cost compared to standard NPO?
>
> **Response**: Thank you for this insightful question. Our approach actually offers advantages in both computational and memory efficiency compared to standard NPO: The reverse policy $1-\pi_\theta$ is automatically derived from $\pi_\theta$ and immediately available at each training step, which does not require  manually crafting or updating a separate reference model. It also eliminates the need to store a frozen reference model $\pi_\text{ref}$ as required by NPO.
>
> ---
> **Question**: Can CaTNiP be composed sequentially for multiple unlearning tasks (e.g., forgetting multiple domains)?
>
> **Response**: Thank you for this insightful question about extending CATNIP to continual unlearning scenarios. Continual unlearning presents distinct challenges beyond single-task settings and typically requires specialized architectural strategies, such as orthogonal adapter parameterization [1] and inter-task relation analysis [2]. Our current work focuses on achieving effective single-task unlearning, which can serve as a building block for such systems.
>
> We conducted preliminary experiments on sequential unlearning across multiple benchmarks. We observed a common phenomenon shared by all single-task unlearning methods evaluated in this paper: utility degradation occurs when these objectives are naively applied in sequence.  We believe that addressing continual unlearning requires joint efforts in data curation, model adapter design, objective formulation, and evaluation protocols, which can be beyond the scope of our current work. However, we consider combining our method with these orthogonal efforts in continual unlearning frameworks an intriguing future direction.
>
>
> [1] Gao, Chongyang, et al. "On large language model continual unlearning." ICLR 2025
>
> [2] Wuerkaixi, Abudukelimu, et al. "Adaptive localization of knowledge negation for continual llm unlearning." ICML 2025

---

> ### Author Response · Authors · 2025-11-24
> **Response to Reviewer JFzo**
>
> **Question**: Interpretability: Could the authors provide visualizations of token-level unlearning beyond the single case study (Fig. 2)?
>
>
> **Response**: Thank you for this insightful question regarding token-level unlearning visualizations. We expanded the case study in Figure 2 by analyzing additional tokens across two manually labeled categories: (1) Harry Potter-related tokens as representative "sensitive" tokens, and (2) non-sensitive tokens related to grammatical and syntactic patterns.  For each group, we report average token probabilities before and after unlearning under the base model, baseline method (NPO), and CATNIP.  $\Delta \%$ indicates the percentage of token probability changes compared with base model.
>
> Results are summarized in Table 1 (Harry Potter-related tokens) and Table 2 (non-sensitive token). We have two key observations: 1) tokens containing sensitive information are  unlearned faster with CATNIP (achieving higher token probability drop $\Delta$ than NPO), while 2) Grammatical and syntactic tokens largely maintain their probabilities. These analysis validated CATNIP's ability to achieve fine-grained, targeted unlearning.
>
> **Table 1**: Model Probability Changes in Harry Potter-related Tokens.
> | Sensitive Token   | Base model | ΔNPO (\%)    | ΔCATNIP (\%) |
> |---------|------------|----------|----------|
> | phoenix | 0.6714     | 10.04   | -62.04  |
> | McG (McGonagall)     | 0.9981     | 0.12   | -48.43  |
> | erva (Minerva)   | 0.9998     | -0.01  | -30.04  |
> | Tom (Tom Riddle)    | 0.8990     | -30.66  | -63.32  |
> | oldemort (Voldemort)| 0.9964     | -100  | -96.34  |
> | Az (Azkaban)     | 0.9876     | -51.04  | -15.73  |
> | G (Goblet of Fire)      | 0.8782     | 2.47   | -41.84  |
> | Harry   | 0.9858     | -3.18  | -25.64  |
> | Ron     | 0.9940     | 0.03   | -26.54  |
> | Ruf (Rufus)     | 0.0888     | -99.12  | -91.12  |
> | Sc (Scabber)     | 0.9691     | -30.45  | -75.06  |
> | Snape   | 0.9975     | -0.59  | -90.07  |
> | Viktor (Viktor Krum)  | 0.8799     | -4.67  | -67.69  |
> | magical | 0.5552     | -13.41  | -66.21  |
>
> **Table 2**: Model Probability Changes in Non-sensitive Tokens.
> | Non-sensitive Token | Base model | ΔNPO(\%) | ΔCATNIP (\%)|
> |-------|------------|------------|------------|
> | 's | 0.8079 | -10.29 | -12.00 |
> | all | 0.9999 | 0.01 | -6.28 |
> | in | 0.9999 | 0.00 | -0.31 |
> | let | 0.9995 | 0.02 | -1.23 |
> | on | 0.9999 | 0.00 | -2.83 |
> | our | 0.9996 | 0.03 | -3.08 |
> | us | 0.9996 | 0.01 | -16.42 |
> | We | 0.9982 | 0.13 | -8.12 |
> | a | 0.3049 | -52.76 | -51.64 |
> | as | 0.9570 | 1.36 | -14.21 |
> | by | 0.9981 | 0.15 | -3.06 |
> | name | 0.3423 | -3.16 | -11.54 |
> | pet | 0.9990 | 0.04 | -4.51 |
> | school | 0.9989 | 0.05 | -5.81 |
> | the | 0.5781 | -37.63 | -15.78 |
>
> ---
> We are running an additional experiment on 7B+-scale LLMs to further validate the generality of our findings. Due to computational cost, this experiment is still in progress. Importantly, all existing evidence presented in the rebuttal has strongly supported our conclusions, and the preliminary results we have obtained so far are consistent with the reported trends.
>
> ---
>
> Once again, we sincerely appreciate you for the thoughtful feedback and constructive questions. We look forward to any further feedback you may have.
>
> Sincerely,\
> Authors

---

### Official Review · Reviewer_agwH · 2025-10-31

**Soundness:** 3
**Presentation:** 4
**Contribution:** 3
**Rating:** 6
**Confidence:** 3

**Summary:**

This work proposes a loss objective (CATNIP) for LLM unlearning. The proposed loss is motivated by negative preference optimization (NPO) but differs in two aspects that address the reference model bias and the token-level bias. Concretely, CATNIP uses $1-\pi_\theta(\cdot|x)$ as the reference model, which corresponds to the model’s confidence in its output. In addition, CATNIP performs a token-wise average instead of a response-wise average to allow for varying confidence levels within the same response. Notably, CATNIP achieves comparable (or better) performance to NPO and SimNPO on multiple unlearning benchmarks, including MUSE and WMDP, without using additional retention data.

**Strengths:**

I find the paper well-written and easy to understand. The motivation of CAINIP is explained. The authors also perform extensive evaluation and comparison of their method with other baselines. I find the experimental results convincing and demonstrate the effectiveness of CAINIP.

**Weaknesses:**

It is unclear why $1-\pi_\theta(\cdot|x)$ is a good choice for reference model (or confidence). In principle any positive function decreasing in  $\pi_\theta(\cdot|x)$ can be used as a reference model.

**Questions:**

1. As above, what are the motivations for using $1-\pi_\theta(\cdot|x)$ as the reference model.


2.  I wonder whether the authors have observed different unlearning speeds across tokens within the same unlearned samples. More specifically, do tokens containing sensitive information tend to be unlearned faster, while tokens providing non-sensitive information (e.g., grammatical structure) maintain higher probabilities?


3. Can the performance of CATNIP be further improved if adding a retention loss as in NPO+KL?

---

> ### Author Response · Authors · 2025-11-24
> **Response to Reviewer agwH**
>
> Dear reviewer agwH,
>
> Thank you for your valuable comments on our paper. We sincerely appreciate your acknowledgment of the CAINIP's effectiveness and the overall quality of our work. Below are our responses to address the concerns you raised.
>
> ---
> **W1 & Q1**. Unclear why using $1 - \pi_\theta$ is a good choice for the reference model. In principle, any positive function decreasing in $\pi_\theta(\cdot|x)$  can be used as a reference model.
>
> **Response**: Thank you for raising this important question and for engaging with the details of our objective.  While we acknowledge that any positive function decreasing in $\pi_\theta$ could theoretically serve as a reference model, our choice of $1-\pi_\theta$ offers three key advantages:
>
> **(1)** It effectively forms  a *counterfactual*  guidance from the start. In the initial unlearning stage, the target policy $\pi_\theta$ is prone to generating forgetting knowledge. Using $1-\pi_\theta$ is more likely to refuse to generate such content,  providing a strong initial guidance. In contrast, $\pi_\text{ref} = \pi_\theta$ serves as a weaker reference when training starts, as it mirrors rather than counters the models' undesirable behavior.
>
> **(2)** During unlearning, $1 - \pi_\theta$ is automatically adapted and immediately available, which **naturally derives** a token weight $w_i$ (Eq 9.) There is no need to manually craft a confidence weight as other work does (*e.g.* WGA [1]), or storing a static reference from a frozen model copy,  thus saving more runtime memory.
>
> **(3)**  Our choice of reference model also smoothes the confidence weight and stablizes training (In our experiments, we always used $\beta >=1$):
>
> - As shown in Figure 1 of the paper, given $\beta >= 1$, the weight $w_i(\beta, \pi)$ is bounded. Especially, when $\beta>1$, the weight function gets a smooth curve when the current policy $\pi_\theta$ is over-confident in generating or refusing  a token( *i.e.*  $\pi_\theta \to0$ or $\pi_\theta \to 1$).
> - When $\pi_\theta$ reaches maximum uncertainty ( $\pi_\theta \to 0.5$), the weight derivative reaches the highest slope, which can appropriately amplify the learning signal.
>
>
>
> Additionally, experiments comparing with [1][2] (as suggested by another Reviewer prd3) show that, although alternative weight functions that decrease with $\pi_\theta(\cdot|x)$ can achieve reasonable unlearning, our methods outperform them in overall performance, supported by good properties as discussed above.
>
> [1] Rethinking LLM Unlearning Objectives: A Gradient Perspective and Go Beyond. ICLR2025.
> [2] Exploring Criteria of Loss Reweighting to Enhance LLM Unlearning. ICML2025.
>
> ---
>
> **Q2**. I wonder whether the authors have observed different unlearning speeds across tokens within the same unlearned samples. More specifically, do tokens containing sensitive information tend to be unlearned faster, while tokens providing non-sensitive information (e.g., grammatical structure) maintain higher probabilities?
>
> **Response**: Thank you for this insightful question regarding token-level unlearning behavior. We expanded the case study in Figure 2 by analyzing additional tokens across two manually labeled categories: (1) Harry Potter-related tokens as representative "sensitive" tokens, and (2) non-sensitive tokens related to grammatical and syntactic patterns.  For each group, we report average token probabilities before and after unlearning under the base model, baseline method (NPO), and CATNIP.  $\Delta \%$ indicates the percentage of token probability changes compared with base model.
>
> Results are summarized in Table 1 (Harry Potter-related tokens) and Table 2 (non-sensitive token). We have two key observations: 1) tokens containing sensitive information are indeed unlearned faster with CATNIP (achieving higher token probability drop $\Delta$ than NPO), while 2) Grammatical and syntactic tokens largely maintain their probabilities. These analyses validated CATNIP's ability to achieve fine-grained, targeted unlearning.

---

> ### Author Response · Authors · 2025-11-24
> **Response to Reviewer agwH**
>
> **Table 1**: Model Probability Changes in Harry Potter-related Tokens.
> | Sensitive Token   | Base model | ΔNPO (\%)    | ΔCATNIP (\%) |
> |---------|------------|----------|----------|
> | phoenix | 0.6714     | 10.04   | -62.04  |
> | McG (McGonagall)     | 0.9981     | 0.12   | -48.43  |
> | erva (Minerva)   | 0.9998     | -0.01  | -30.04  |
> | Tom (Tom Riddle)    | 0.8990     | -30.66  | -63.32  |
> | oldemort (Voldemort)| 0.9964     | -100  | -96.34  |
> | Az (Azkaban)     | 0.9876     | -51.04  | -15.73  |
> | G (Goblet of Fire)      | 0.8782     | 2.47   | -41.84  |
> | Harry   | 0.9858     | -3.18  | -25.64  |
> | Ron     | 0.9940     | 0.03   | -26.54  |
> | Ruf (Rufus)     | 0.0888     | -99.12  | -91.12  |
> | Sc (Scabber)     | 0.9691     | -30.45  | -75.06  |
> | Snape   | 0.9975     | -0.59  | -90.07  |
> | Viktor (Viktor Krum)  | 0.8799     | -4.67  | -67.69  |
> | magical | 0.5552     | -13.41  | -66.21  |
>
> **Table 2**: Model Probability Changes in Non-sensitive Tokens.
> | Non-sensitive Token | Base model | ΔNPO(\%) | ΔCATNIP (\%)|
> |-------|------------|------------|------------|
> | 's | 0.8079 | -10.29 | -12.00 |
> | all | 0.9999 | 0.01 | -6.28 |
> | in | 0.9999 | 0.00 | -0.31 |
> | let | 0.9995 | 0.02 | -1.23 |
> | on | 0.9999 | 0.00 | -2.83 |
> | our | 0.9996 | 0.03 | -3.08 |
> | us | 0.9996 | 0.01 | -16.42 |
> | We | 0.9982 | 0.13 | -8.12 |
> | a | 0.3049 | -52.76 | -51.64 |
> | as | 0.9570 | 1.36 | -14.21 |
> | by | 0.9981 | 0.15 | -3.06 |
> | name | 0.3423 | -3.16 | -11.54 |
> | pet | 0.9990 | 0.04 | -4.51 |
> | school | 0.9989 | 0.05 | -5.81 |
> | the | 0.5781 | -37.63 | -15.78 |
>
> **Q3**. Can the performance of CATNIP be further improved if adding a retention loss as in NPO+KL?
>
> **Response**: Thank you for this valuable question. While CATNIP is designed to work effectively without retention data, it becomes even more powerful when such data is available. We conducted experiments combining CATNIP with a KL retention loss, with results summarized in  **Table 3**.  CATNIP + KL achives the best tradeoff in unlearning and utility preservation. In contrast, GA + KL largely hurts unlearning efficacy, while removing the KL retention term causes catastrophic forgetting. This indicates CATNIP exhibits minimal interference between unlearning and retention objectives, and demonstrates higher compatibility with retention constraints.
>
> **Table3:** Evaluating CATNIP with retention loss incoporated.
> | Method                         | Know f ↓ (Extended) | Δf ↓ (Extended) | Know f ↓ (MUSE) | Δf ↓ (MUSE) | MMLU ↑ | Δu ↑ | ΔO ↑ |
> |--------------------------------|---------------------|------------------|------------------|--------------|--------|-------|-------|
> | **Base model**                 | 39.99               | -                | 32.13            | -            | 60.45  | -     | -     |
> | GA + KL (w/ $D_r$)               | 38.29               | (✗)              | 27.20            | (✓)          | **60.18** | (✓)     | 1.43  |
> | NPO + KL (w/ $D_r$)              | 33.62               | (✗)              | 28.92            | (✗)          | 59.47  | (✓)     | 5.39  |
> | FLAT (w/ $D_{ct}$)                 | 5.44                | (✓)              | 6.35             | (✓)          | 50.12  | (✗)   | 24.22 |
> | **CATNIP (Ours) + KL (w/ $D_r$)**| **0.00**            | (✓)              | **0.00**         | (✓)          | 59.48  | (✓)     | **39.02** |
> | GA                             | **0.00**            | (✓)              | **0.00**         | (✓)          | 24.87  | (✗)     | -5.61 |
> | NPO                            | 25.21               | (✗)              | 24.18            | (✗)          | 54.79  | (✓)  | 9.12  |
> | SimNPO                         | 6.87                | (✓)              | 6.54             | (✓)          | 51.84  | (✓)     | 24.21 |
> | **CATNIP (Ours)**              | **2.29**            | (✓)              | **2.08**         | (✓)          | 52.17  | (✓)     | **29.42** |
>
> Once again, we sincerely appreciate you for the thoughtful feedback and constructive questions. We look forward to any further feedback you may have.
>
> Sincerely,\
> Authors

---

### Author Response · Authors · 2025-12-03
**Summary of Rebuttal and Revisions**

Dear ACs, SACs, and PCs,

Thank you for overseeing the review process, and we thank all reviewers for their careful evaluation and constructive feedback on our submission. To assist the newly assigned AC, we provide below a concise summary of the key review points and how we addressed them during the author–reviewer discussion period.

Reviewers broadly agreed that CaTNiP is a  principled negative-preference–based unlearning method, which (1) offers a new thoretical framework grounded on  preference-ranking of dynamic reverse policies,  (2) achieves strong forget–retain trade-offs on three tasks including MUSE-books,  WMDP-Bio, WMDP-Cyber, and (3) works effectively in a retention-data-free regime.

**Efforts to address reviewer concerns**: During the author-reviewer discussion period, we  took substantial additional work to address the reviewers' comments. Especially,


* **Comparison between tokenized unlearning methods**. Following reviewer prd3's suggestions, we added WGA [1] and SatImp [2] as baselines. As summarized in Tables 1 & 2, our method is consistently competitive on  unlearning benchmarks including WMDP-Cyber, WMDP-Bio, and MUSE-Book. WGA shows non-negligible MMLU
(↑) drops across both WMDP-Bio and Cyber tasks. SatImp mitigates the over-forgetting issue on WMDP-Cyber while notably underperforming on WMDP-Bio. Overall, CaTNiP demonstrates the strongest trade-off between unlearning effectiveness and utility preservation using only the undesirable forgetting data. Moreover, in Section 5.2, Figure 4, we analyze the gradient-weighting effects of these tokenized unlearning methods, providing additional empirical interpretation of CaTNiP’s gains in addition to its theoretical grounding.

* **Performance on TOFU benchmark**. Following reviewer prd3's suggestions, we expanded experiments on the TOFU benchmark which  showed that our method is comparative or outperforming recent tokenized unlearning methods (WGA and SatImp) in terms of forgetting and utility preservation (MU’). Detailed results are updated in Section A.8 of our paper.

* **Unlearning performance comparison when retention data is available**. We applied the commonly adopted KL retention loss to different unlearning methods. As shown in the updated Table 2 of the paper, most retention-free unlearning methods, including NPO and WGA, exhibit a non-negligible drop in forgetting quality. The retention regularization cannot improve the utility preservation of SatImp. In contrast, our method  preserves utility without compromising unlearning. This suggests that CaTNiP exhibits minimal interference between unlearning and retention objectives and demonstrates higher compatibility with explicit retention constraints.


* **Performance on larger models**. In response, we compared performance of retention-free methods on a larger scale model, Qwen7B-Instruct (compared to LLama 3.2 3B-Instruct),  and observed that all methods except CaTNiP exhibit a notable drop in forgetting quality when applied to larger models, highlighting the consistent unlearning efficacy of CaTNiP across model scales (see Table 13).


* We provided more detailed, token-wise case studies which validated CaTNiP's ability to achieve fine-grained, targeted unlearning in forgetting-domain-specific tokens while preserving general syntactic patterns (Updated in Section A.7).



In summary, we have provided extensive new experiments, analyses, and revisions that directly address the reviewers’ main concerns and further support the paper’s core conclusions. We respectfully hope that, in light of these rebuttal efforts, the work will receive your favorable consideration.

Sincerely,\
Authors

---
[1] Rethinking LLM Unlearning Objectives: A Gradient Perspective and Go Beyond. ICLR2025.

[2] Exploring Criteria of Loss Reweighting to Enhance LLM Unlearning. ICML2025.

---

### Meta-Review · Area_Chair_w6nn · 2026-01-03

**Summary:**

The paper introduces a new LLM unlearning method called CaTNiP (Calibrated and Tokenized Negative Preference Alignment). CaTNiP addresses failure modes of prior methods like Gradient Ascent, in particular when it comes to (i) their sample-wise optimization limitation, making prior methods pay more attention to longer samples, and (ii) static reference point in prior preference-optimization-based approaches which limits the effectiveness of unlearning some samples during some phases of optimization.
CaTNiP addresses the first issue by operating at a token-level rather than the "response-" or "sample"-level. It addresses the second issue by using a dynamic "reverse policy" as a reference. This leads to the unlearning update for a given token to be scaled based on the model's confidence.
A notable feature of CaTNiP also is that it does not rely on "retention data".

Several reviewers found the paper well-written and easy to understand (Reviewer agwH, Reviewer prd3). Reviewer JFzo appreciated the clarity of mathematical derivations. The reviewers also found the motivation of the work to be well-stated and justified (Reviewer agwH, Reviewer prd3, Reviewer JFzo). Reviewers (e.g. Reviewre vLqU) appreciated the theoretical foundation of the work "deriving the method from preference optimization principles and providing clear gradient formulations that explain why it outperforms GA or NPO, backed by ablation studies that isolate components like tokenization and calibration".

The reviewer raised the following main concerns:
**C1**. Limited justification for the choice of reference model (Reviewer agwH), limited theoretical explanation for why the proposed method leads to stable optimization (Reviewer JFzo).
**C2**. Lacking comparisons with recent baselines that perform localized editing/unlearning (Reviewer JFzo).
**C3**. Evaluation limited to mid-size models (Reviewer JFzo, Reviewer prd3, Reviewer vLqU) and limited from the perspective of not including all relevant datasets (Reviewer prd3, vLqU)
**C4**. Lack of novelty due to highly related works that were not cited, and then (after long exchanges with the authors and additional experimental results), lack of performance improvement above those related works (Reviewer prd3).

**Reviewer Concerns:**

The largest outstanding concern relates to the lack of novelty of the work relative to previously-published works that the authors had not cited in their original submission, and lack of comprehensive comparisons against these works. While the authors have cited these works and compared against them empirically in the rebuttal and in their updated paper revision, these preliminary results have some drawbacks and are not complete (see more details below), warranting further empirical investigation to accurately frame the contributions of this work relative to prior published studies.

Other unresolved concerns relate to the scalability of the method, with several reviewers pointing out that the authors evaluated only mid-size models. While the authors claim that they are running larger-scale experiments during the rebuttal, new results were not reported on this.

Other concerns relating to elaborating on the justification of different design choices were well addressed during the rebuttal.

**Reviewer Scores:**

**Reviewer agwH**.
This reviewer was overall positive about the paper. The only weakness pointed out is a missing justification for why the specific function is chosen as a reference model. The authors have addressed this well by providing extensive justification and updating the paper to include it.
The authors have also engaged thoroughly with each other question that the reviewer raised, about speed of unlearning of different tokens, and compatibility of CaTNiP with a utility preservation loss, running new experiments for each, which all yielded interesting findings: sensitive tokens are unlearned faster than non-sensitive ones, and CaTNiP’s performance can further be boosted with the inclusion of a retention loss.

**Reviewer JFzo**.
The authors provided additional intuition and justification behind their choice of reference model and its role in stabilizing training dynamics. But they did not offer theoretical guarantees for this.  The authors claimed to be running larger-scale experiments but did not have the chance to present the results. Further, the concern about lacking comparisons to recent localized unlearning baselines has not been addressed.

**Reviewer prd3**.
This reviewer raised the most serious concerns about the paper, strongly advocating that it does not meet the bar for acceptance. The main argument is that the paper lacks highly relevant references, the inclusion of which significantly limits the novelty of the paper, making it incremental.
The authors argued against the incremental nature of their work, claiming that, despite their method having a similar flavour as other previous works (in terms of gradient reweighting and token-level objective), they have presented a principled framework for arriving at their proposed method, rather than using heuristics.
The view of the reviewer, which remained unchanged throughout the discussions, is that the reasoning for how the specific mechanism was obtained is irrelevant if that mechanism itself resembles others that are already in existence and yields incremental performance gains over them.
The reviewer and authors had several interactions discussing experimental details, making it evident that the performance of the relevant prior baselines that the reviewer pointed out can vary significantly depending on the particular evaluation setup and hyperparameter configuration of these methods, which in turn makes it hard to draw a clear conclusion about whether the new method truly outperforms those prior works or is simply “competitive” with them.
In the end, the reviewer remains unconvinced about the novelty of the paper and its empirical superiority over prior works and expresses doubts about the validity of the results that the authors obtained, since the reviewer was able to obtain better results with the prior methods in question than the results that the authors presented (due to using different hyperparameters).


**Reviewer vLqU**
This reviewer is overall positive about the paper. They raised concerns about the reference policy, similarly to other reviewers, which I believe the authors have addressed sufficiently. They also raised concerns about excluding relevant datasets for evaluation, and about limited evidence of scalability, due to evaluating on medium sized models only. The reviewer also asked the authors how their contribution relates to the same recent works that were pointed out by Reviewer prd3. The concern about the scalability remains unaddressed, and the other concerns relating to datasets and baselines have been partially addressed via a rebuttal that echoes the arguments that the authors offered to Reviewer prd3.
Had there been a discussion phase, Reviewer VLqU (whose confidence was 3) may have been convinced by Reviewer prd3 (whose confidence was 5) that the size of contribution of this paper is smaller than originally seemed, due to the existence of the related works that both of these reviewers brought up.

**Overall**, while the paper certainly has merits (including the theoretical framework provided, strong empirical performance on some benchmarks, the retain-free nature of CaTNiP, and excellent writing quality, among others), I recommend rejection of the paper in its current form as I feel that a further revision is necessary to present the contributions of the paper accurately in light of other relevant published works, as well as to comprehensively compare against those prior methods. Given the heavy reliance of some of those methods on hyperparameter settings (and the complexity of unlearning evaluations, with many important trade-offs at play), more experiments should be conducted in a careful way (beyond what was possible in the short rebuttal phase) to fairly and comprehensively compare these methods against one another.

---

### Decision · Program_Chairs · 2026-01-26

Reject